# Impaired Myocardial Mitochondrial Function in an Experimental Model of Anaphylactic Shock

**DOI:** 10.3390/biology11050730

**Published:** 2022-05-10

**Authors:** Walid Oulehri, Olivier Collange, Charles Tacquard, Abdelouahab Bellou, Julien Graff, Anne-Laure Charles, Bernard Geny, Paul-Michel Mertes

**Affiliations:** 1Pôle Anesthésie, Réanimation Chirurgicale, Hôpitaux Universitaires de Strasbourg, 67091 Strasbourg, France; walid.oulehri@chru-strasbourg.fr (W.O.); olivier.collange@chru-strasbourg.fr (O.C.); charlesambroise.tacquard@chru-strasbourg.fr (C.T.); 2Faculté de Médecine de Strasbourg, UR 3072 Institut de Physiologie, FMTS (Fédération de Médecine Translationnelle de Strasbourg), Université de Strasbourg, 67091 Strasbourg, France; anne.laure.charles@unistra.fr (A.-L.C.); bernard.geny@chru-strasbourg.fr (B.G.); 3Institute of Sciences in Emergency Medicine, Academy of Medical Sciences, Guangdong General People Hospital, Guangzhou 510060, China; abellou402@gmail.com; 4Department of Emergency Medicine, Wayne State University School of Medicine, Detroit, MI 48201, USA; 5Global Healthcare Network & Research Innovation Institute LLC, Brookline, MA 02446, USA; 6Faculté de Médecine de Strasbourg, Institut d’Histologie, Service Central de Microscopie Électronique, FMTS (Fédération de Médecine Translationnelle de Strasbourg), Université de Strasbourg, 67091 Strasbourg, France; julien.graff@unistra.fr; 7Service de Physiologie et d’Explorations Fonctionnelles, Pôle de Pathologie Thoracique, Hôpitaux Universitaires de Strasbourg, 67091 Strasbourg, France

**Keywords:** anaphylactic shock, cardiac dysfunction, mitochondria, lipid peroxidation, reactive oxygen species, peroxynitrite, SOD, electronic microscopy

## Abstract

**Simple Summary:**

Anaphylactic shock (AS) is the most severe allergic hypersensitivity reaction. In the early stage of shock, AS was associated with hemodynamic impairments: severe failure in arterial blood pressure and cardiac dysfunction. Mitochondrial disorders are associated with a high incidence of acute or chronic cardiac dysfunctions. The aim of the study was to evaluate mitochondrial involvement in cardiac dysfunction at the early stage of AS. In control (CON) and sensitized Brown Norway rats, anaphylactic shock (AS) was induced with ovalbumin iv bolus injection. Aortic abdominal blood flow (ABF), mean blood arterial pressure (MAP), and lactatemia were assessed during 15 min. The myocardial mitochondrial defect in AS was assessed by mitochondrial respiration, oxidative stress production, total superoxide dismutase activity (SODs), and oxidative damage. An ultrastructural evaluation of myocardial mitochondria was also performed using electronic microscopy. AS was associated with a rapid and dramatic decrease in MAP, ABF, and severe hyperlactatemia 15 min after AS induction. Non-phosphorylating mitochondrial respiration and OXPHOS states involving mitochondrial complex II were significantly impaired in the AS group. Oxidative stress was observed with a significant increase in SODs activity after AS induction. Moreover, in the AS group we observed an increase in oxidative damage via lipid peroxidation. No modifications of the mitochondrial ultrastructure were shown at the early stage of AS. In this experimental model, AS was associated with a rapid and significant myocardial mitochondrial dysfunction with oxidative damage that could explain cardiac anaphylaxis dysfunction.

**Abstract:**

Anaphylactic shock (AS) is associated with a profound vasodilation and cardiac dysfunction. The cellular mechanisms underlying AS-related cardiac dysfunction are unknown. We hypothesized that myocardial mitochondrial dysfunction may be associated with AS cardiac dysfunction. In controls and sensitized Brown Norway rats, shock was induced by ovalbumin i.v bolus, and abdominal aortic blood flow (ABF), systemic mean arterial pressure (MAP), and lactatemia were measured for 15 min. Myocardial mitochondrial function was assessed with the evaluation of mitochondrial respiration, oxidative stress production by reactive oxygen species (ROS), reactive nitrogen species (RNS), and the measurement of superoxide dismutases (SODs) activity. Oxidative damage was assessed by lipid peroxidation. The mitochondrial ultrastructure was assessed using transmission electronic microscopy. AS was associated with a dramatic drop in ABF and MAP combined with a severe hyperlactatemia 15 min after shock induction. CI-linked substrate state (197 ± 21 vs. 144 ± 21 pmol/s/mg, *p* < 0.05), OXPHOS activity by complexes I and II (411 ± 47 vs. 246 ± 33 pmol/s/mg, *p* < 0.05), and OXPHOS activity through complex II (316 ± 40 vs. 203 ± 28 pmol/s/mg, *p* < 0.05) were significantly impaired. ROS and RNS production was not significantly increased, but SODs activity was significantly higher in the AS group (11.15 ± 1.02 vs. 15.50 ± 1.40 U/mL/mg protein, *p* = 0.02). Finally, cardiac lipid peroxidation was significantly increased in the AS group (8.50 ± 0.67 vs. 12.17 ± 1.44 µM/mg protein, *p* < 0.05). No obvious changes were observed in the mitochondrial ultrastructure between CON and AS groups. Our experimental model of AS results in rapid and deleterious hemodynamic effects and was associated with a myocardial mitochondrial dysfunction with oxidative damage and without mitochondrial ultrastructural injury.

## 1. Introduction

Anaphylactic shock (AS) is a rare but potentially life-threatening immediate allergic hypersensitivity reaction. AS is still associated with significant mortality, even when it is rapidly diagnosed and treated [1].

The classical pathway of anaphylaxis involves an acute and massive allergen-induced release of mediators from tissue mast cells and blood basophils. Both pre-formed and de novo synthesized mediators have deleterious effects in hemodynamics as severe arterial hypotension due to excessive vasodilatation [2]. Genuine cardiac dysfunction has been described during AS [3,4]. Our group has previously shown that AS was associated with early myocardial dysfunction characterized by severe decreases in mean arterial pressure (MAP, −75%), left ventricular inotropy (max dP/dt, −83%), and left ventricular shortening fraction (LVSF, −33%) [5].

The crucial role of myocardial mitochondria has been widely reported in acute and chronic heart failure [6], and the mechanisms underlying heart disease-related mitochondrial dysfunction are well described [7,8]. A recent state of the art described molecular mechanisms as mitochondrial dysfunction underlying myocardial ischemic injury [9]. Mitochondrial dysfunction is a major cause of oxidative stress and oxidative damage leading to cell toxicity and death. Indeed, excessive reactive oxygen species (ROS) or nitrogen species (RNS) can lead to oxidative damage to proteins, lipids, and nucleic acids [10]. Mitochondria dysfunction and oxidative stress are particularly sensitive to cells with high levels of energy consumption as cardiomyocytes [11]. Few studies report the contribution of mitochondrial dysfunction and oxidative stress in pathophysiological subcellular mechanisms of anaphylaxis. Moreover, mitochondrial ultrastructure changes have never been studied at the early stage of AS while hemodynamics are severely affected. Mitochondria appear to play a role in anaphylaxis-related pathological mechanisms [12,13] without a clear explanation of which signaling pathways are impaired (mitochondrial respiration, oxidative stress generation, or oxidative damage occurrence). Simultaneously, progress has been made in recent years with the development of mitochondrial-targeted therapy [14] for the treatment of heart diseases. Mitochondrial dysfunctions are involved in many cardiomyopathies. Most important, therapeutics approaches investigated in the last decade for targeting cardiac mitochondria were synthetic mitochondrial antioxidants and mitochondria-targeting natural compounds [6]. 

To our knowledge, the role played by myocardial mitochondria during AS has never been studied.

The objective of our study was to test the hypothesis that AS is associated with myocardial mitochondrial dysfunction related to mitochondrial respiration impairment, oxidative stress production, and oxidative damage.

## 2. Materials and Methods

Fifty six 10 week-old male Brown Norway rats weighting 250–300 g (Janvier, Le Genest-St-Isle, France) were sensitized by subcutaneous administration of chicken egg albumin as previously described [15,16]. Animals were randomized between control (CON, *n* = 28) and shocked (AS, *n* = 28) groups. Hemodynamics were measured in all animals (*n* = 56). 

Twenty eight animals were used to study mitochondrial respiration and ROS production; 14 animals were used to study superoxide dismutase (SOD) activity and reactive nitrogen species (RNS) production; and the 14 remaining rats were used to study oxidative damage, in the CON group (*n* = 7) and in the AS group (*n* = 7) for each experimentation. 

AS was induced by an intravenous injection of 1 mg ovalbumin diluted in 1 mL of 0.9% saline solution (T0). In the control group, 1 mL of normal saline solution was injected (T0). 

All animal procedures and care were performed in accordance with the European Communities Council Directive of 24 November 1986 (86/609/EEC). The study protocol was approved by the Ethical Committee for Animal Experiments of Strasbourg University, France (N° #16987-2018100500047648). 

### 2.1. Surgical Preparation

Surgery began on day 21 following the initial sensitization as previously described [15]. Anesthesia was induced and maintained using isoflurane. The trachea was intubated, and lungs were mechanically ventilated (Alpha Lab, Equipement Vétérinaire Minerve) to maintain PaCO_2_ between 35 and 40 mmHg during the stabilization period. A fluid-filled polyethylene catheter was inserted into the right femoral vein for fluid maintenance (10 mL/kg/h of 0.9% saline solution) and administration of ovalbumin in the shocked group. 

At the end of the experiment and after blood and tissue sampling (T15), rats in both groups were euthanized by an anesthetic overdose.

### 2.2. Hemodynamic Measurements 

Abdominal aortic blood flow (ABF) was measured using perivascular ultrasonic flow probes (Transonic System Inc, Ithaca, New York, NY, USA) placed on the abdominal aorta. Mean arterial blood pressure (MAP) was continuously monitored with a fluid-filled polyethylene catheter inserted into the right femoral artery. Values were recorded prior to shock induction (T0) and then at regular intervals: T1, T2.5, T5, T7.5, T10, T12.5, and T15 min corresponding to the end of the experiment. Arterial blood gas and blood lactate levels were measured before shock induction and at the end of the experiment.

### 2.3. Myocardial Mitochondrial Respiration

At T15 min, the left ventricle was extracted. The mitochondrial respiratory chain was assessed with a high-resolution respirometer (Oxygraph-2 k; Oroboros Instruments, Innsbruck, Austria) with continuous stirring, at 37 °C in a buffer. Briefly, as previously described [17], fibers (1 mg wet weight) were separated and then permeabilized in a bath of solution containing 50 μg/mL saponin for 30 min at 4 °C with shaking. A multiple substrate-inhibitor titration protocol was used for analysis of non-phosphorylating respiration and oxidative phosphorylation. Five mM glutamate and 2 mM malate were used to determine non-phosphorylating respiration initially supported by complex I (CI-linked substrate state). The second step was the injection of succinate (25 mM), which activates complex II to obtain the CI + II-linked substrate state. Then, 2 mM ADP was added at a saturating concentration, which activated the ATP synthase, and oxidative phosphorylation was stimulated through complexes I and II to obtain the maximum OXPHOS coupling state or CI + II-linked OXPHOS state. Complex I was blocked with amytal (0.02 mM) to give the OXPHOS state initiated only through complex II (CII-linked OXPHOS state). Finally, we used antimycin A (2.5 µM) to inhibit complex III and block oxygen consumption of the respiratory chain. We evaluated the residual oxygen consumption (ROX) state due to oxidative sides reactions. 

The oxygen consumption was analyzed using DatLab software 4.3. These data are expressed as pmol O_2_/s/mg wet weight.

### 2.4. Oxidative Stress Measurements

#### 2.4.1. Production of Reactive Oxygen Species (ROS) and Reactive Nitrogen Species (RNS)

ROS were detected by electron paramagnetic resonance (EPR) spectroscopy. A 1-hydroxy-3-methoxycarbonyl-2,2,5,5-tetramethyl-pyrrolidine spin probe (CMH; oxidized form CM, Noxygen^®^, Elzach, Germany) is oxidized when ROS, and especially superoxide anion (O_2_^•−^), is produced [18]. After the EPR measurement, muscles pieces were dried for 15 min at 150 °C, and ROS production was expressed as µmol/min/mg dry weight.

To assess oxidative modifications on nitric oxide (NO), the RNS detected was peroxynitrite (ONOO-), a spontaneously formed superoxide and nitric oxide (NO) product. Peroxynitrite is a strong oxidizing species and highly active nitrating agent. It was measured using a peroxynitrite assay kit (DAX-J2 PON Green) (AAT Bioquest, Sunnyvale, CA, USA). Peroxynitrite production was expressed as relative fluorescence unit (RFU)/mg protein.

#### 2.4.2. Superoxide Dismutases (SODs) Activity

Enzymatic antioxidant superoxide dismutases catalyze the dismutation of superoxide anion into oxygen and hydrogen peroxide. This class of enzymes is an important antioxidant defence in nearly all cells exposed to superoxide radical. SODs activity was detected using the Amplite Colorimetric Superoxide Dismutase Assay Kit (AAT Bioquest, Sunnyvale, CA, USA). SODs activity was expressed as U/mL/mg protein.

### 2.5. Oxidative Damage: Lipid Peroxidation Measurement

Lipid peroxidation was quantified by measuring the concentration of thiobarbituric acid reactive substances (TBARS) using a Quantichrom TBARS assay kit (DTBA-100) (Bioassay System, Hayward, CA, USA). TBARS production was expressed in µM/mg protein.

### 2.6. Bradford Method to Determine Protein Concentration (mg Protein)

The Bradford method [19] was used to determine protein concentration using a spectrophotometer. Bovine serum albumin (BSA) was taken as the standard. Different concentrations of BSA were prepared for the preparation of the standard curve of BSA. One hundred μL of each dilution of BSA was taken in test tubes and labeled according to the concentration. One hundred μL of distilled water was taken in the test tube as a blank. Then, 5 mL of Bradford reagent was added to each test tube, mixed, and placed at room temperature for 15 min. After a few minutes, the sample color changed to blue, and then the absorbance of the tubes was noted at 595 nm. The graph for the standard curve was plotted, taking concentration along the *x*-axis and absorbance along the *y*-axis.

### 2.7. Transmission Electronic Microscopy (TEM)

Left ventricle tissue was cut into 1 mm^3^ small pieces and fixed in 2.5% glutaraldehyde for 48 h. The samples were post fixed in 1% osmic tetroxide for 2 h at 4 °C. The tissues were then dehydrated through an acetone series (30%, 50%, 70%, 90%, and 100%) for 15 min each. The dehydrated tissues were then cleared off the acetone by propylene oxide for 30 min and then infiltrated and embedded in a liquid resin epoxy. After embedding, the samples were prepared by professional person of the Histology Institute of Strasbourg University. Specimens were examined and photographed for mitochondrial ultrastructural changes by TEM Philips EM 208, USA.

### 2.8. Statistical Analysis

Values were expressed as mean ± standard error of the mean (SEM). Parameters were compared between groups using the Student t-test. Gaussian distributions were assessed by using Shapiro–Wilk’s test. A *p*-value < 0.05 was considered as statistically significant. Statistical analyses were performed using Prism 6.0 (Graph Pad Software Inc., San Diego, CA, USA).

## 3. Results

### 3.1. Hemodynamics during AS

AS induced a prompt and dramatic decrease in ABF and MAP (Figure 1A,B); arterial hyperlactatemia (1.5 ± 0.3 mmol/L vs. 5.8 ± 0.4 mmol/L, *p* < 0.05; respectively in CON and AS groups), metabolic acidosis (pH 7.41 ± 0.02 vs. 7.20 ± 0.03 and HCO_3_- 23.6 ± 0.3 vs. 8.9 ± 0.7 mmol/L, *p*< 0.05; in CON and AS groups, respectively), and hypocapnia (PaCO_2_ 38 ± 1 vs. 23 ± 1 mmHg, *p* < 0.05; in CON and AS groups, respectively).

### 3.2. Myocardial Mitochondrial Respiration and Oxidative Stress in AS

Activities of the respiratory chain complexes are presented in Figure 2.

There was no significant difference in non-phosphorylating respiration with activation of complex I alone (CI-linked substrate state) between CON and AS groups, respectively: 124 ± 16 vs. 93 ± 15 pmol/s/mg, *p* = 0.18. However, myocardial mitochondrial CI + II-linked substrate state was significantly decreased in the AS group: CI + II-linked substrate state (−27%, 197 ± 21 vs. 144 ± 21 pmol/s/mg, *p* < 0.05; in CON and AS groups, respectively).

AS also impaired the maximal OXPHOS state as shown by the CI + II-linked OXPHOS activity (−40%, 411 ± 47 vs. 246 ± 33 pmol/s/mg, *p* < 0.05; in CON and AS groups, respectively) and the OXPHOS state only through complex II: CII-linked OXPHOS activity (−36%, 316 ± 40 vs. 203 ± 28 pmol/s/mg, *p* < 0.05; in CON and AS groups, respectively). Figure 3 illustrates oxygen flux between groups depending on substrates or inhibitors added in Oxygraph chambers with the corresponding mitochondrial respiration states.

Concerning ROX state evaluation, we did not observe statistical difference among groups: 15 ± 8 vs. 16 ± 7 pmol/s/mg, *p* = 0.67 in CON and AS groups, respectively.

### 3.3. Oxidative Stress Measurement

#### 3.3.1. ROS and RNS Production 

ROS production (Figure 4A) was not significantly different in CON vs. AS group respectively (0.14 ± 0.01 vs. 0.17 ± 0.03 µmol/min/mg dry weight, *p* = 0.62). Concerning RNS (Figure 4B), peroxynitrite production was not significantly different in CON vs. AS group, respectively (0.87 ± 0.08 vs. 1.15 ± 0.13 RFU/mg protein, *p* = 0.07).

#### 3.3.2. SODs Activity

SODs activity (Figure 4C) was significantly higher in AS group (11.15 ± 1.02 vs. 15.50 ± 1.40 U/mL/mg protein, *p* = 0.02; in CON and AS group, respectively).

### 3.4. Oxidative Damage with Lipid Peroxidation

TBARs production (Figure 4D) was significantly increased in AS group (8.50 ± 0.67 vs. 12.17 ± 1.44 µM/mg protein, *p* = 0.04; in CON and AS group, respectively).

### 3.5. Electron Microscopy Findings to Myocardial Mitochondria

Cardiac mitochondria were observed after AS via TEM to assess modifications at the microstructure level. As shown in Figure 5, no obvious changes were found in CON group and in AS group 15 min after AS onset. Mitochondria were slender with round shapes. The mitochondrial membrane remains clear and intact, and inside their membranes, long and homogenous cristae were arranged orderly and densely.

## 4. Discussion

The major findings of the present study are summarized as follows: our AS model was associated with a rapid myocardial mitochondrial dysfunction characterized by a −30 to −40% significant decrease in respiratory chain complex activity and a significant lipid peroxidation damage. 

In this study, we used an experimental model of AS widely described in the literature. As we demonstrated previously [15,20,21,22], those studies confirm our results using 1 mg of OVA.

Hemodynamic alterations confirmed those previously described using the same experimental model [5,15]. The rapidity and severity of shock may be a direct consequence of the massive release of mediators with an immediate action on cardiac function and systemic hemodynamics [23,24]. Our model is characterized by a decrease in cardiac output with decrease of nearly 90% of ABF, a profound arterial hypotension with a decrease of nearly 80% of MAP, hypocapnia, and hyperlactatemia. Hemodynamic and metabolic changes occurred 15 min after AS induction. Explosive reactions are common in AS but unusual in other types of shock (e.g., septic shock) where hemodynamic manifestations only appear once compensatory physiological reactions have been exhausted. Myocardial dysfunction during AS has also been described in animals with a dramatic reduction in inotropism [5] and in humans where elevated troponin I, left ventricular (LV) systolic dysfunction, and abnormal regional LV wall motion were observed [25]. While ventilation parameters remained constant in the two groups, hypocapnia reflects the decrease in regional blood flow and highlights the damage induced by AS at the tissue or cellular level [2]. The cellular mechanisms associated with severe hemodynamic dysfunction in AS have not been elucidated yet. 

Our results suggest that AS is associated with an inhibition of mitochondrial complex II activity with an increase in peroxynitrite and lipid peroxidation in cardiomyocytes. We used saponin-skinned fibers that ensure global mitochondrial functional assessment in intact mitochondria (not isolated and potentially fragilized mitochondria). Hence, we evaluated oxygen consumption by other cell fractions, and no difference was observed in the ROX state that highlights the poor involvement of other cell fractions in oxygen consumption in our AS model. In the present study, all mitochondrial respiration states involving complex II, when using succinate or rotenone supporting that complex II activity was increased, were impaired in AS group. Mitochondrial involvement in acute and chronic cardiac dysfunction as myocardial infarction [26], toxic cardiomyopathy [27], or arrhythmias [28] is well established. Similarly to our results, shock-induced complex II impairment has been demonstrated by Gellerich et al. in an experimental model of septic shock: cardiac dysfunction was associated with a significant decrease in complex II [29]. More recently, Jain-Ghai et al. reported that a mutation in complex II is associated with cardiac dysfunction in human [30]. Probable explanations of this early and selective mitochondrial chain impairment are directly related to various post-translational modifications of acetylation or dephosphorylation subunits [31]. After AS, a possible signaling cascade that would modulate subunits and regulate complex II activity remain to be uncovered. Interestingly, in an indirect way, complex II activity can be modulated by peroxynitrite (ONOO-) or superoxide (O_2_^•−^). In this study, we observed a clear trend towards higher peroxynitrite production (+25%, *p* = 0.07). This nitrative specie, as ROS, could rapidly interact with mitochondrial complex II and modulate its activity. Zhang et al. proposed in a myocardial infarction model a specific profile of complex II subunits oxidative modifications mediated by peroxynitrite and protein tyrosine nitration [32]. Similarly, complex II-induced superoxide production has a deleterious effect by inducing self-inactivation through oxidative damage of the protein matrix of complex II [33]. This interaction may explain in part that significant differences were not observed between CON and AS groups in RNS/ROS production within 15 min. 

By itself, peroxynitrite can modulate mitochondrial complex II activity and damage a wide array of biomolecules including lipids and eventually contributing to the cell death [34]. Indeed, Beckman et al. identified peroxynitrite as the source of cellular damage by membrane lipid peroxidation [35,36]. Overall, ROS/RNS produced by dysfunctional mitochondria are well known to rapidly oxidize lipid components and therefore cause lipid peroxidation [37,38]. This rapid interaction may also explain in part the lack of significant differences between both groups in ROS/RNS production.

Herein, we observed oxidative damage within 15 min of AS related to lipid peroxidation. Cardiomyocytes cell damage could be another source of cardiac dysfunction observed in our model. Horton et al. noted that lipid peroxidation contributes to myocardial dysfunction in a model of ischemia reperfusion [39]. 

The role of peroxynitrite should be considered as crucial in AS. This reactive species is formed from the reaction between superoxide and nitric oxide (NO). NO is the principal vasodilatator involved in AS and is produced by nitric oxide synthase (NOS) [40]. The constitutive endothelial isoform (eNOS) is particularly involved in NO production probably through histamine-induced eNOS phosphorylation, which enhances enzymatic activity [41]. 

We did not observe a significant increase in ROS production. Although, this finding seems contradictory to the increase in RNS. A low ROS level could be explained by, first, rapid interaction between ROS and downstream intracellular molecules such as NO. Second, ROS production may be difficult to measure. In our study, we used the spin trapping method to detect free radicals by EPR, which is the most unambiguous method for the detection of free radicals [42]. However, in some cases, this method remains limited in some biological systems [43]. Indeed, spin trapping of many free radicals proceeds with much lower rates as compared with their scavenging with cellular antioxidants [44]. For example, superoxide dismutase (SOD) and ascorbate may prevent superoxide radical detection in cells and tissues by preventing O_2_^•−^ detection [45]. Moreover, these species rapidly interact with intracellular molecules and therefore have a very short half-life, making it more difficult to demonstrate an increase in oxidative stress.

We observed a significant increase in SOD in the AS group at T15 min. In this study, the increase in SOD activity may be a consequence of two cell signaling pathways: (1) a prompt excessive accumulation in ROS during AS that enhanced enzymatic activity to counteract the ROS burden as SOD increased is considered as protective system and (2) Ca^++^ dependent signaling processes in the mitochondrial matrix [46]. An increase in SOD activity could reflect an earlier increase in oxidative stress within the first 15 min. Further studies evaluating oxidative stress at 5 and 10 min after AS onset would highlight the early stage of AS pathophysiology. Few studies described the role of antioxidant enzymes such as SOD during AS, but Lee et al. reported that treatment with human recombinant extracellular SOD inhibits airway inflammation in the ovalbumin-induced mice allergy model of [47].

Surprisingly, electronic microscopy analysis showed a normal shape of cardiomyocytes mitochondria without alteration of cristae and normal mitochondrial membrane. This finding demonstrates that alteration of the mitochondrial respiration chain we observed in our AS model is probably not due to mitochondrial structural lesions caused by AS and hemodynamic defects.

Our model of AS is characterized by mitochondrial respiration impairment associated with a deleterious oxidative damage by lipid peroxidation and at least via the peroxynitrite pathway (Figure 6). Our findings suggest that anti-oxidative treatment should be tested during experimental AS. For example, N-acetylcysteine (NAC) is a thiol that can directly reduce radicals by donating one electron [48] with the ability to react with various reactive compounds as ROS or RNS [49]. Experimental studies have shown the effectiveness of NAC in improving cardiac function with a cardioprotective role in conditions such as acute myocardial infarction, heart failure, or coronary artery disease [50].

This study presents some limitations. The results presented in this work should be confirmed by in vitro or genetic experiments to demonstrate a potential causal relationship between cardiac mitochondrial dysfunction and AS. However, such approaches have significant limitations, including that the short time frame of AS (15 min) might not allow the detection of significant molecular pathways changes. We did not repeat the measurement of maximum dP/dt previously assessed on the same model [5] that would have supported more accurate evaluation of cardiac function. Indeed, the measurement of maximum dP/dt requires the insertion of a probe into the left heart ventricle that can induce myocardial injuries, thus biasing the evaluation of cellular lesions directly related to the AS itself. Although our study proposes a possible role of functional mitochondrial defect induced by AS, generalization to human AS should be taken cautiously. Further studies are needed to determine whether mitochondrial protection and antioxidants might limit cardiac dysfunction after AS. 

## 5. Conclusions

In conclusion, our findings reveal a new intracellular mechanism involved in a severe AS rat model associated with impairment of myocardial mitochondrial respiration and increased lipid peroxidation without ultrastructural injury.

## Figures and Tables

**Figure 1 biology-11-00730-f001:**
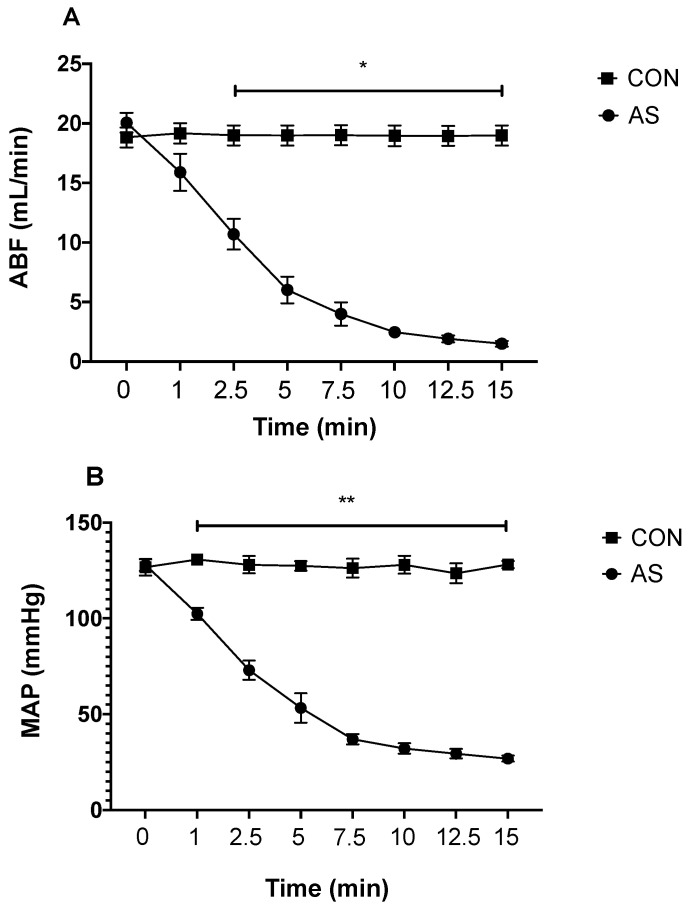
Hemodynamic parameters. Time-course of hemodynamic parameters in control (CON, *n* = 42) and anaphylactic shock (AS, *n* = 42) groups: (**A**) abdominal aortic blood flow (ABF), (**B**) systemic mean arterial pressure (MAP). T0: time of ovalbumin injection. Values are means ± SEM. * The difference in ABF was significantly different between the CON and AS groups at each measurement time (*p* < 0.01). ** The difference in MAP was significantly different between the CON and AS groups at each measurement time (*p* < 0.01).

**Figure 2 biology-11-00730-f002:**
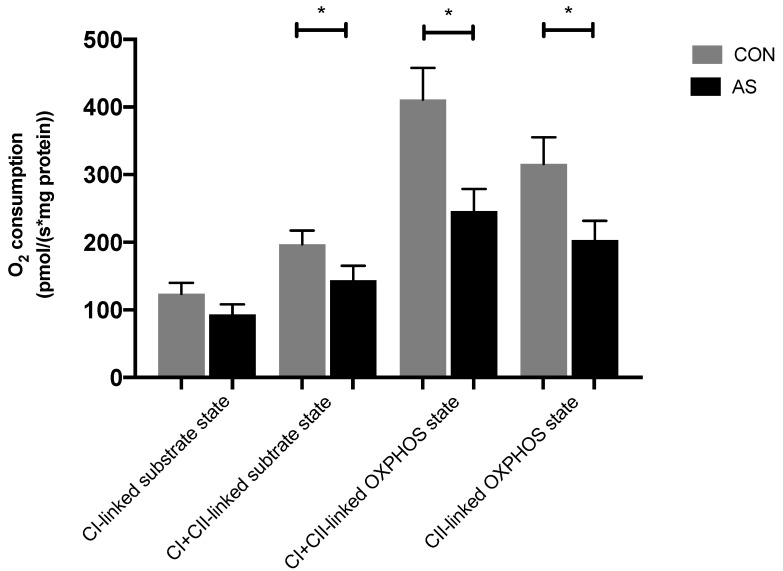
Oxidative capacity and activity of respiratory chain complexes in cardiac muscle in CON compared with AS group. CON: control group; AS: anaphylactic shock group; CI: mitochondrial complex I; CI + II: mitochondrial complexes I and II; OXPHOS: mitochondria ADP-activated state of oxidative phosphorylation, CII: mitochondrial complex II. Values are means ± SEM, * *p* < 0.05.

**Figure 3 biology-11-00730-f003:**
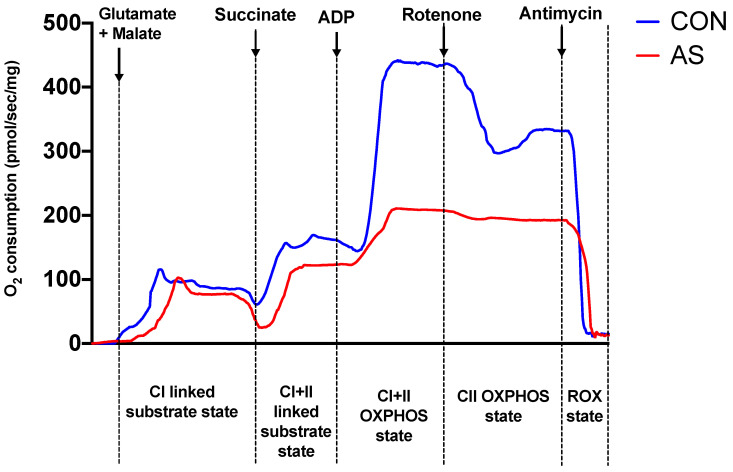
Representative respirometry trace from permeabilized cardiac muscle fibers in the Oxygraph-2 K chambers. Oxygen consumption flux are shown from each chamber. Blue curve indicates oxygen flux in chamber A (control group, CON), and red curve indicates oxygen flux in chamber B (shocked group, AS). Both chambers operated simultaneously. Substrates and inhibitors were sequentially added to each chamber (arrows) to evaluate activity of mitochondrial complexes. Representative areas of each oxygen flux were separated to indicate mitochondroal respiration state (below the figure). CI: complex I; CI + II: complexes I and II; CII: complex II; OXPHOS: oxidative phosphorylation; ROX state: residual oxygen consumption state.

**Figure 4 biology-11-00730-f004:**
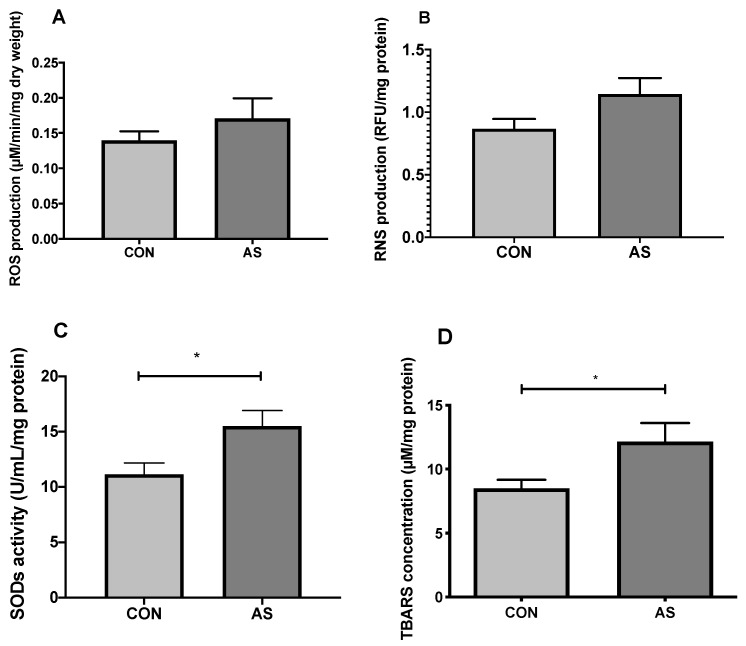
Oxidative stress and damage after AS. Myocardial mitochondrial oxidative stress 15 min after AS onset with ROS production (**A**) was not significantly increased (*p* = 0.62), and RNS production (**B**) tended to increase (*p* = 0.07) in AS group. SODs activity (**C**) was significantly increased in AS group, which might explain the low rate of ROS in AS group. TBARs concentration (**D**) was significantly increased in AS group probably due to RNS-pathway with peroxynitrite. Results were expressed as mean ± SEM. * *p* < 0.05 compared to CON group. CON: control group, AS: shock group, ROS: reactive oxygen species, RNS: reactive nitrogen species, SODs: superoxide dismutases.

**Figure 5 biology-11-00730-f005:**
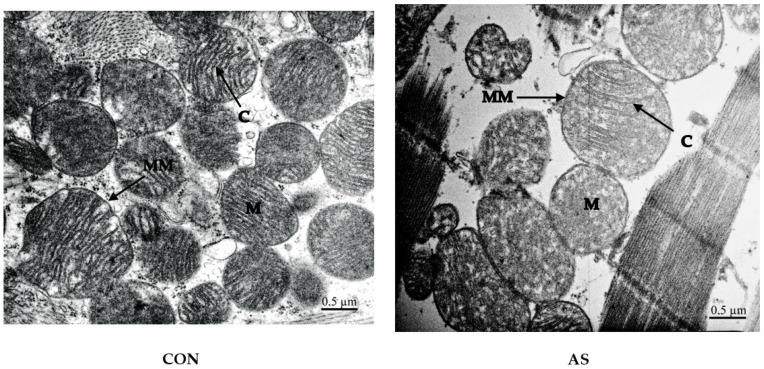
Representative TEM images of cardiomyocytes (×20k magnification) in CON and AS groups. No obvious changes were observed between groups in mitochondrial shapes (M), mitochondrial membrane (MM), and cristae (C).

**Figure 6 biology-11-00730-f006:**
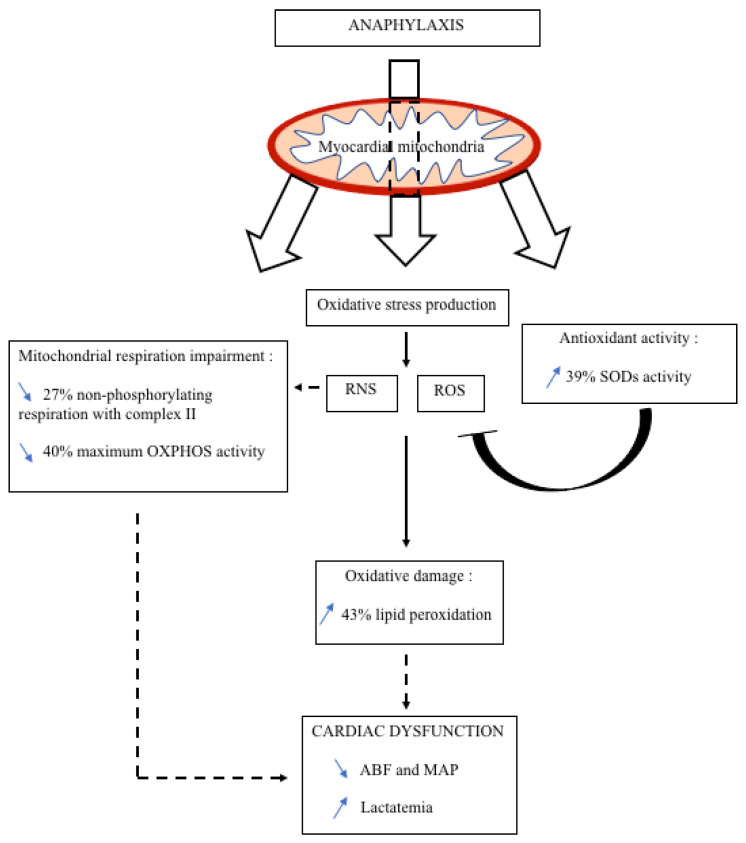
Illustration of anaphylaxis–suspected pathophysiological mechanisms involving myocardial mitochondrial dysfunction leading at least in part of cardiac dysfunction described in AS. In this study, AS leads to mitochondrial respiration impairment at T15 min in non-phosphorylating respiration with complex II and in maximum OXPHOS activity, which might alter cardiac function. AS-induced oxidative stress leads to oxidative damage with an increase in lipid peroxidation that might similarly impair cardiac function probably through NO-pathway, and its nitrative species as peroxynitrite, and less through ROS because of scavenging by the increase in SODs activity. Complete arrows: relationship based on results of this study. Dotted arrows: relationships based on the literature.

## Data Availability

Not applicable.

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
