# Peer review of "Impaired Myocardial Mitochondrial Function in an Experimental Model of Anaphylactic Shock"

_biology, 2022, doi:10.3390/biology11050730_

Round 1

Reviewer 2 Report

In this manuscript, the authors have investigated myocardial mitochondrial function in rats with anaphylactic shock (AS). Although this study would be important for understanding the relationship between AS and impaired myocardial mitochondria, I don't think that the present set of data are conclusive enough to draw unambiguous structural conclusions. I think this manuscript has some serious problems as indicated below.

  1. The fundamental problem is the lack of data to support the illustration hypothesis in Fig. 4. In this in vivo experiment, administration of ovalbumin to rats only caused mitochondrial and cardiac dysfunction, but their relevance is unclear. I doubt that in AS state, not only myocardial mitochondria but that in other parts of the body may also be impaired. Throughout this in vivo experiment, the causal relationship of each result is uncertain.
  2. Since the results are only numerical data (Fig. 1-3), they are not reliable. Imaging results (such as immunohistochemistry and electron microscopic findings to myocardial mitochondria, etc.) should be presented to make the conclusions more convincing.
  3. Insufficient verification of AS. Since there is little verification regarding mast cells and basophils, it is unclear whether this experimental rat model correctly represents the pathogenesis of AS. The comparison between only two group (the 1 mg dose of ovalbumin and the normal group) is also poorly investigated. As a result, this manuscript is no more than a report that administration of ovalbumin to rats causes abnormalities in various parts of the body (including myocardial mitochondria).

The experimental model itself is good, but I think that the results are not well discussed and conclusions are not meaningful in this manuscript. Without the results of in vitro experiments or genetic experiments in this study, the causal relationship between each result was not pursued. In the present state, this manuscript is inappropriate for publication in journals with a high impact factor.

Round 2

Reviewer 1 Report

Thanks to the authors for the work done to improve the quality of the article.

Reviewer 2 Report

Thank you for the opportunity to review this paper again. I have previously determined it to be "Reject", but the authors have responded to the reviewers' comments sincerely and adequately. Although I do not wholly agree with the authors' research model, the idea should be published because this revised manuscript presents a massive effort.